# The N-Terminal Region of the Polo Kinase Cdc5 Is Required for Downregulation of the Meiotic Recombination Checkpoint

**DOI:** 10.3390/cells10102561

**Published:** 2021-09-27

**Authors:** Sara González-Arranz, Isabel Acosta, Jesús A. Carballo, Beatriz Santos, Pedro A. San-Segundo

**Affiliations:** 1Instituto de Biología Funcional y Genómica (IBFG), Consejo Superior de Investigaciones Científicas (CSIC) and University of Salamanca, 37007 Salamanca, Spain; biologa@usal.es (S.G.-A.); isacosta@usal.es (I.A.); bsr@usal.es (B.S.); 2Department of Cellular and Molecular Biology, Centro de Investigaciones Biológicas Margarita Salas, Consejo Superior de Investigaciones Científicas (CSIC), 28040 Madrid, Spain; j.carballo@cib.csic.es; 3Departamento de Microbiología y Genética, University of Salamanca, 37007 Salamanca, Spain

**Keywords:** meiosis, meiotic recombination checkpoint, Cdc5, polo kinase, yeast

## Abstract

During meiosis, the budding yeast polo-like kinase Cdc5 is a crucial driver of the prophase I to meiosis I (G2/M) transition. The meiotic recombination checkpoint restrains cell cycle progression in response to defective recombination to ensure proper distribution of intact chromosomes to the gametes. This checkpoint detects unrepaired DSBs and initiates a signaling cascade that ultimately inhibits Ndt80, a transcription factor required for *CDC5* gene expression. Previous work revealed that overexpression of *CDC5* partially alleviates the checkpoint-imposed meiotic delay in the synaptonemal complex-defective *zip1Δ* mutant. Here, we show that overproduction of a Cdc5 version (Cdc5-ΔN70), lacking the N-terminal region required for targeted degradation of the protein by the APC/C complex, fails to relieve the *zip1Δ*-induced meiotic delay, despite being more stable and reaching increased protein levels. However, precise mutation of the consensus motifs for APC/C recognition (D-boxes and KEN) has no effect on Cdc5 stability or function during meiosis. Compared to the *zip1Δ* single mutant, the *zip1Δ cdc5-ΔN70* double mutant exhibits an exacerbated meiotic block and reduced levels of Ndt80 consistent with persistent checkpoint activity. Finally, using a *CDC5*-inducible system, we demonstrate that the N-terminal region of Cdc5 is essential for its checkpoint erasing function. Thus, our results unveil an additional layer of regulation of polo-like kinase function in meiotic cell cycle control.

## 1. Introduction

Evolutionarily conserved Polo-like kinases (PLKs) are essential regulators of both mitotic and meiotic cell cycles. In addition to the kinase domain, PLKs are characterized by the presence of the polo-box domain (Figure 1A) that mediates the localization and recognition of their substrates, often primed by previous phosphorylation, providing an exquisite spatiotemporal mechanism for their regulation [1]. Unlike other organisms, such as worms and mammals, in the budding yeast *Saccharomyces cerevisiae* there is a single member of this PLK family, the Cdc5 protein. Cdc5 is involved in multiple processes during the mitotic cell cycle, including entry into mitosis, cohesion cleavage, mitotic exit, and cytokinesis. In addition, Cdc5 also participates in the DNA damage response [2].

During the specialized meiotic cell cycle, a series of highly regulated events must be accomplished to ensure the distribution of exactly half of the chromosome complement to the gametes. These requirements include the establishment of physical connections between homologous chromosomes via pairing, synapsis, and recombination during prophase I, sequential release of sister-chromatid cohesion at chromosome arms, and centromeric regions during meiosis I and II, respectively, and mono-orientation of sister kinetochores during the first meiotic segregation [3,4,5]. Cdc5 performs paramount roles in all these processes, highlighting the relevance of PLKs also during meiosis [6]. Cdc5 is required for exit from prophase I, triggering synaptonemal complex (SC) destruction and resolution of recombination intermediates [7,8,9]. Next, Cdc5 is also essential for meiotic chromosome segregation promoting removal of centromeric cohesion by Rec8 phosphorylation and establishing coorientation of sister kinetochores by triggering the nucleolar release of Lsr4/Csm1 and monopolin association to kinetochores [10,11,12,13].

Like in mitosis, cell-cycle surveillance mechanisms or checkpoints also operate during meiosis to ensure accuracy in the completion of critical events. In particular, the meiotic recombination checkpoint blocks or delays the prophase I to meiosis I transition in response to unrepaired DNA double-strand breaks (DSBs) arising from defects in synapsis or recombination [14]. Briefly, unrepaired resected DSBs are initially detected by the Mec1-Ddc2 complex [15], followed by the phosphorylation of Hop1, a component of the lateral elements (LEs) of the SC, by the activated Mec1 kinase. Red1, another component of the LEs is an essential mediator of this phosphorylation event. Among the different residues of Hop1 that can be potentially targeted by Mec1, phosphorylation of Hop1-T318 is critical for the checkpoint response [16,17,18,19]. The action of the Pch2 ATPase is also critical to sustain adequate levels of Hop1-T318 phosphorylation and Hop1 localization to axes [20,21,22], thus relaying the checkpoint signal through the activation of the Mek1 effector kinase [23]. Different targets of Mek1 have been identified, including Hed1-T40 and Rad54-T132 that contribute to preventing intersister recombination by different mechanisms [24,25]. Mek1 also phosphorylates threonine 11 of histone H3 (H3-T11ph) [26]. Although the biological relevance of this phosphorylation remains elusive, it is a useful readout for checkpoint-induced Mek1 activity [27]. Finally, Mek1 also phosphorylates and inhibits Ndt80, which constitutes a crucial downstream target of the meiotic checkpoint signaling pathway [28]. Ndt80 is a transcription factor that activates the expression of a number of genes including those encoding factors required to execute the prophase I to meiosis I transition, such as the cyclin Clb1 and the polo-like kinase Cdc5. The inhibition of *CLB1* transcription upon checkpoint-dependent inactivation of Ndt80 by Mek1 is thought to contribute to meiotic cell cycle arrest together with the inhibitory phosphorylation of CDK1 (Cdc28) by Swe1 [29,30]. Cdc5, in turn, is involved in a feed-forward mechanism that results in checkpoint inactivation by promoting Red1 degradation [31], thus enabling exit from prophase I when DSBs are repaired and the checkpoint is satisfied.

The levels of the Cdc5 protein during the mitotic cell cycle are exquisitely regulated. After exiting from mitosis, Cdc5 is degraded during G1 by the action of the APC/C-Cdh1 complex [32,33]. Deletion of the first 70 amino acids of Cdc5 containing the regulatory motifs, D-boxes, recognized by APC/C greatly stabilizes the protein [33]. Failure to degrade Cdc5 in the mutant lacking the N-terminal region (hereafter, *cdc5-ΔN70*) has detrimental consequences for the cell cycle, including impaired control of Cdc14 nucleolar release [34]. During meiosis, Cdc5 protein stability is also controlled by APC/C, but in this case, associated with the meiosis-specific activator Ama1 [35]. Ama1-dependent control of the levels of Cdc5 and other factors enables the extension of meiotic prophase I length compared to the mitotic G2 phase [36].

In this work, we examine the relevance of the N-terminal domain of Cdc5 for the regulation of the activity of the meiotic recombination checkpoint. We show that the N-terminal region comprising the first 70 amino acids of Cdc5 encompassing the destruction box is required for the ability of the polo kinase to override meiotic checkpoint signaling. Unexpectedly, we found that this requirement is independent of the consensus motifs recognized by APC/C, which are also dispensable for controlling Cdc5 protein stability during meiosis. Likewise, the regulation of the checkpoint by the N-terminal domain of Cdc5 does not require known sites of CDK-dependent phosphorylation. Thus, our results uncover potentially unrecognized aspects of Cdc5 function and regulation that impinge on the control of meiotic checkpoint activity.

## 2. Materials and Methods

### 2.1. Plasmids

Plasmids used in this work are listed in Appendix A. Plasmid pJC29 containing wild-type *CDC5* with an HA epitope at the start codon in the high-copy vector pRS426 was described in [37] and kindly provided by David Morgan and Sue Jaspersen. The pSS122 plasmid overexpressing *cdc5-ΔN70* was constructed as follows. A DNA fragment lacking the sequence encoding the first 70 amino acids of Cdc5 was generated by fusion PCR of two fragments with overlapping sequences amplified using pJC29 as a template. The fusion fragment was digested with *Cla*I and used to replace the *Cla*I-*Cla*I fragment of pJC29. Plasmid pSS360 overexpressing *cdc5-db7A* contains an *Xho*I-*Bam*HI fragment from pSS359 (see below) in pRS426.

Plasmids pSS308, pSS309, pSS310, and pSS311, containing *cdc5-T23A*, *cdc5-T70A*, *cdc5-S18A*, and *cdc5-T29A*, respectively, were constructed by cloning a 2.7 kb *Eco*RI-*Bam*HI fragment from pEM130, pEM131, pEM137, and pEM138, respectively, into the same sites of pRS426. Plasmids pEM130, pEM131, pEM137, and pEM138 have been described in [38], and they were kindly provided by D. Kellog.

Plasmids pSS252 and pSS253 contain an *Xho*I-*Sac*I fragment from pJC29 or pSS122 harboring *CDC5* or *cdc5-ΔN70*, respectively, flanked by promoter and terminator regions, cloned into the same sites of pBluescript SK+. The *natMX4* marker was then inserted at the unique *Xba*I site located in the 3′UTR of *CDC5* and *cdc5-ΔN70* in pSS252 and pSS253 to generate pSS254 and pSS255, respectively. Plasmid pSS359 was generated by replacing the *Eco*RI-*Nsi*I fragment in pSS254 with a synthetic fragment (gBlock, IDT) containing the *cdc5-db7A* mutations. pSS254, pSS255, and pSS359 were used for targeting *CDC5*, *cdc5-ΔN70*, and *cdc5-db7A*, respectively, to the genome after *Xho*I-*Sac*I digestion. 

Plasmid pMJ998 was kindly provided from M. Lichten and contains *P_GAL1_-CDC5*, *P_GPD1_-GAL4*(*1-848-ER*), and the *natMX4* marker that can be targeted to the *CDC5* genomic locus after *Nsi*I digestion. Plasmid pSS425 derives from pMJ998 by deleting the sequence corresponding to the first 70 amino acids of Cdc5 in *P_GAL1_-CDC5*.

### 2.2. Strains

Yeast strains and genotypes are listed in Appendix A. All strains used in this work are in the BR1919 background, except the *cdc5-1* thermosensitive mutant used in Figure 1C, which is W303. The *zip1Δ:LYS2* and *ndt80Δ:LEU2* gene deletions have been previously described. To replace the endogenous *CDC5* gene with the *cdc5-ΔN70* or *cdc5-db7A* mutant versions, wild-type or *zip1Δ* strains were transformed with *Xho*I-*Sal*I-digested pSS255 and pSS359 plasmids (see above) and selected for clonNAT resistance. As a control, the pSS254 plasmid containing the original wild-type *HA-CDC5* was also used. To generate *CDC5-IN*, the *zip1Δ ndt80Δ* double mutant was transformed with pMJ998 cut with *Nsi*I. The integration of this construct results in a non-tandem duplication of *CDC5* with the following configuration: *P_CDC5_-CDC5**—P_GPD1_-GAL4(1-848)ER**—natMX4**—P_GAL1_-CDC5.* In the case of the *cdc5-ΔN70-IN* strain, *Nsi*I-digested pSS425 was used leading to this genomic disposition: *P_CDC5_-CDC5**—P_GPD1_-GAL4(1-848)ER**—natMX4**—P_GAL1_-cdc5-ΔN70*. All constructions and mutations were confirmed by PCR analysis and/or sequencing. The sequences of all primers used in strain construction are available upon request. All strains were made by direct transformation of haploid parents or by genetic crosses always in an isogenic background. Diploids were made by mating the corresponding haploid parents and isolation of zygotes by micromanipulation. The plasmids and strains used in each figure are listed in Appendix A.

### 2.3. Meiotic Time Courses, Sporulation Efficiency and Spore Viability

To induce meiosis and sporulation, BR strains were grown in 3.5 mL of synthetic complete medium (2% glucose, 0.7% yeast nitrogen base without amino acids, 0.05% adenine, and complete supplement mixture from Formedium at twice the particular concentration indicated by the manufacturer) for 20–24 h, then transferred to 2.5 mL of YPDA (1% yeast extract, 2% peptone, 2% glucose, and 0.02% adenine) and incubated to saturation for an additional 8 h. Cells were harvested, washed with 2% potassium acetate (KAc), resuspended into 2% KAc (10 mL), and incubated at 30 °C with vigorous shaking to induce meiosis. Both YPDA and 2% KAc were supplemented with 20 mM adenine and 10 mM uracil. The culture volumes were scaled up when needed. To induce *CDC5* or *cdc5-ΔN70* from the *P_GAL1_* promoter in strains expressing *GAL4-ER (CDC5-IN* or *cdc5-ΔN70-IN* strains), 1 μM β-estradiol (Sigma E2257, St. Louis, MO, USA) dissolved in ethanol was added to the cultures 24 h after meiotic induction in prophase-arrested *zip1Δ ndt80Δ* cells (Appendix A).

To score meiotic nuclear divisions, samples from meiotic cultures were taken at different time points, fixed in 70% ethanol, washed in phosphate-buffered saline (PBS), and stained with 1 μg/μL 4′,6-diamidino-2-phenylindole (DAPI) for 15 min. At least 300 cells were counted at each time point. Meiotic time courses were repeated several times. Averages and error bars from at least three counts are shown.

Sporulation efficiency was quantified by microscopic examination of asci formation after 62 h on a liquid sporulation medium. Both mature and immature asci were scored. Spore viability was determined by dissection of spores from asci and assessing the ability to form colonies after 3 days of incubation at 30 °C on YPDA plates.

### 2.4. Microscopy

To analyze meiotic nuclear divisions in DAPI-stained cells, a Leica DMRXA fluorescence microscope equipped with a Hamamatsu Orca-AG CCD camera and a 63X 1.4 NA objective was used. Bright-field microscopy images of asci were captured with a Nikon Eclipse 90i microscope controlled with MetaMorph software and equipped with a Hamamatsu Orca-AG CCD camera and a PlanApo VC 63X 1.4 NA objective.

### 2.5. Western Blotting

Total cell extracts for Western blot analysis were prepared by trichloroacetic acid (TCA) precipitation from 5 mL aliquots of sporulation cultures, as previously described [39]. The antibodies used are listed in Appendix A. The ECL or ECL2 reagents (ThermoFisher Scientific, Waltham, MA, USA) were used for detection. The signal was captured on films or with a Fusion FX6 system (Vilber, Marne-la-Vallée, France) and quantified with the Evolution-Capt software (Vilber).

## 3. Results

### 3.1. The N-Terminus of Cdc5 Is Required to Bypass the zip1Δ-Induced Meiotic Checkpoint

We have previously reported that overexpression of *CDC5* from a high-copy plasmid partially suppresses the robust delay in meiotic progression of the *zip1Δ* mutant imposed by the action of the meiotic recombination checkpoint ([39]; Figure 1B). Experiments using a kinase-dead allele have revealed that Cdc5 kinase activity is required for its effect on meiotic checkpoint bypass [39]. In order to gain additional insight into this meiotic function of Cdc5, we searched for other domains of Cdc5 that could influence its impact on the meiotic recombination checkpoint response.

The N-terminus of Cdc5 has been previously described to be important for its function during mitosis. This region harbors a so-called destruction box that contains motifs recognized by the APC/C complex targeting Cdc5 for degradation at the end of mitosis [33]. We decided to investigate whether this domain is important for Cdc5 meiotic checkpoint function. We constructed a multicopy plasmid expressing a version of *CDC5* lacking the sequence corresponding to the first 70 amino acids harboring the destruction box (*cdc5-ΔN70*; Figure 1A). We transformed a *zip1Δ* mutant strain with 2μ-based high-copy plasmids expressing full-length *CDC5*, the *cdc5-ΔN70* version, or the empty vector, and monitored the kinetics of meiotic progression by DAPI staining of nuclei. A wild-type strain transformed with an empty vector was also included as a reference for normal meiotic progression. As expected, the *zip1Δ* mutant transformed with vector alone displayed a strong meiotic arrest, undergoing meiotic divisions inefficiently and only after prolonged incubation under sporulation conditions (Figure 1B) [29]. Overexpression of *CDC5* in *zip1Δ* resulted in an earlier onset of meiotic nuclear divisions and more efficient meiotic progression, though below wild-type levels, as described ([39]; Figure 1B). In contrast, overexpression of *cdc5-ΔN70* in the *zip1Δ* strain had no effect on the delayed and inefficient meiotic progression of *zip1Δ* (Figure 1B). The fact that *cdc5-ΔN70* overexpression does not suppress *zip1Δ* meiotic delay is not due to a lack of Cdc5 kinase activity of this truncated construct because, like the plasmid expressing the wild-type version, it was able to complement the growth defect conferred by the thermosensitive *cdc5-1* allele at the restrictive temperature (Figure 1C). We also monitored by Western blot the levels of Cdc5 and Ndt80 in these meiotic cultures. As expected, and consistent with the kinetics or meiotic divisions, the production of Ndt80 and Cdc5 was significantly delayed in *zip1Δ* compared to the wild type (Figure 1D). According to previous observations [39], in the *zip1Δ* mutant transformed with the multicopy plasmid expressing wild-type *CDC5*, the presence of the Cdc5 protein was detected at earlier time points, leading to an accelerated induction of Ndt80 production (at t = 20 h) in comparison with the *zip1Δ* strain transformed with an empty vector (at t = 38 h). In contrast, although overexpression of the *cdc5-ΔN70* version led to sustained levels of this truncated form, it did not alter the deferred onset of Ndt80 induction characteristic of *zip1Δ* (t = 38 h). Notably, the delayed appearance of Ndt80 paralleled the late induction of the endogenous full-length Cdc5 protein expressed from the genomic locus (Figure 1D). Collectively, these results indicate that this Cdc5 N-terminal protein domain (1–70 amino acids) is essential for the ability of Cdc5 to suppress the checkpoint-induced meiotic delay of *zip1Δ* when overproduced.

### 3.2. The Meiotic Checkpoint Function of Cdc5 N-Terminal Domain Is Independent of the Consensus Motifs for Targeted Degradation via APC/C

The 1–70 N-terminal domain of Cdc5 contains two RXXL “D-box” motifs and a “KEN” motif (Figure 1A). These signaling sequences are present in proteins whose APC/C-dependent degradation is tightly orchestrated at specific cell-cycle stages [40,41]; in the case of Cdc5, it is targeted for degradation by the APC/C^Cdh1^ complex at the end of mitosis [32,33]. Indeed, at least the KEN motif in the N-terminal domain of Cdc5 has been shown to be functionally relevant to control the stability of a chimeric Cdc5^N1−80^-GFP protein in mitotic cells [42]. Since we observed that deletion of the 1–70 N-terminal of Cdc5 prevented its capacity to suppress the *zip1Δ*-induced meiotic checkpoint response, we examined whether this outcome was due to the lack of signaling from the D-box and KEN motifs, which are absent in the *cdc5-ΔN70* construct. To address this possibility, we used site-directed mutagenesis to create a *CDC5* allele (*cdc5-db7A*) in which all conserved amino acids in both RXXL and KEN consensus motifs were mutated to alanine (Figure 1A). Surprisingly, unlike *cdc5-ΔN70*, a high-copy plasmid containing *cdc5-db7A* resulted in similar dynamics of Cdc5-db7A production during meiosis to that conferred by wild-type *CDC5* overexpression in *zip1Δ* cells; moreover, the timing of Ndt80 induction was also similar in both conditions (Figure 1D). Accordingly, overexpression of *cdc5-db7A* partially suppressed the delayed meiotic progression of *zip1Δ* to the same extent as the wild-type *CDC5* did (Figure 1B). We conclude that the functional contribution of the 1–70 N-terminal domain of Cdc5 to the control of the meiotic recombination checkpoint response does not rely on its regulation via the consensus D-box and KEN motifs.

### 3.3. CDK Sites in the N-Terminal Domain of Cdc5 Are not Required for Its Meiotic Checkpoint Function

In addition to the APC/C recognition motifs, the 1–70 N-terminal domain of Cdc5 possesses several sites identified as targets of CDK phosphorylation in mitotic cells (Figure 1A; [38]). A function for some of these sites in the regulation of Cdc5 protein function or stability during the mitotic cell cycle has been reported [43,44,45]. In order to assess whether the inability of *cdc5-ΔN70* overexpression to accelerate the meiotic progression of *zip1Δ* was due to the absence of a particular CDK phosphorylation event in the 1–70 N-terminal region, we generated high-copy plasmids carrying *cdc5-S18A*, *cdc5-T23A*, *cdc5-T29A*, or *cdc5-T70A* phosphomutants, in which individual CDK target sites have been changed to alanine (Figure 1A). Overexpression of either *cdc5-S18A*, *cdc5-T29A*, or *cdc5-T70A* increased the efficiency of meiotic divisions in *zip1Δ* to comparable levels as overexpression of wild-type *CDC5* did (Figure 2A). Likewise, induction of Ndt80 production was similarly accelerated upon overexpression of the S18A, T29A, and T70A mutants (Figure 2B, green arrowheads), paralleling the increased levels of Cdc5 protein achieved at earlier time points (Figure 2B, green asterisks). These results suggest that phosphorylation of S18, T29, and T70 is not required for the ability of Cdc5 overproduction to override the meiotic recombination checkpoint. On the other hand, the high-copy plasmid containing *cdc5-T23A* transformed into *zip1Δ* conferred little or no effect on meiotic divisions (Figure 2A) and Ndt80 production (Figure 2B, red arrowheads), displaying similar levels and kinetics to those of to the *zip1Δ* strain transformed with empty vector. However, we note that, in contrast with the other phosphomutants analyzed, the multicopy plasmid expressing *cdc5-T23A* does not generate higher levels of the Cdc5-T23A protein (Figure 2B, red asterisks), thus explaining its incapability to alleviate *zip1Δ* meiotic delay. Indeed, this result is consistent with the observation that the stability of the Cdc5-T23A protein is compromised [44]. Since, unlike *cdc5-T23A*, overexpression of *cdc5-ΔN70* leads to high steady-state levels of the Cdc5-ΔN70 protein (Figure 1D), we conclude that the inability of Cdc5-ΔN70 to suppress the checkpoint-dependent block of *zip1Δ* does not stem from the absence of CDK phosphorylation events in the known target sites of the Cdc5 N-terminal region.

### 3.4. The N-Terminal Domain of Cdc5, but Not the APC/C Recognition Motifs, Is Required for Efficient Meiotic Progression and Sporulation

We next generated diploid strains in which we replaced the endogenous wild-type *CDC5* gene by either the *cdc5-ΔN70* or *cdc5-db7A* versions at its own genomic locus to further investigate the relevance of the N-terminal domain of Cdc5 for meiotic development. We found that higher levels of the Cdc5-ΔN70 protein were detected throughout the whole meiotic time course (Figure 3A) suggestive of increased stability of the protein lacking the N-terminal domain. The peak of Cdc5-ΔN70 (t = 20–24 h) was coincident with the induction of Ndt80 indicating that its gene expression is also normally controlled by Ndt80-dependent transcriptional regulation. In contrast, the dynamics and levels of Cdc5-db7A production were similar to those of the wild type, indicating again that Cdc5 stability during meiosis is not significantly regulated by canonical APC/C signaling via the known consensus motifs. Furthermore, analysis of the kinetics of meiotic nuclear divisions revealed that meiotic progression was slightly, but reproducibly, slower, and less efficient in the *cdc5-ΔN70* mutant (Figure 3B). However, meiotic progression was not altered in *cdc5-db7A* (Figure 3B). We also monitored the formation of spores by microscopic examination of asci. The *cdc5-ΔN70* mutant, but not *cdc5-db7A*, showed decreased sporulation efficiency (Figure 3C) with a marked reduction in the formation of mature 4-spored asci. In particular, *cdc5-ΔN70* exhibited a preponderance of dyads compared to the wild type or the *cdc5-db7A* mutant (Figure 3D and 3E). The immaturity of *cdc5-ΔN70* asci largely impeded their micromanipulation to assess spore viability, and it was impossible to dissect full 4-spore tetrads. In any case, the subset of spores that could be dissected from the *cdc5-ΔN70* mutant showed reduced viability (Figure 3F). We conclude that the 1–70 N-terminal domain of Cdc5 contains regulatory features different from the RXXL and KEN boxes that contribute to the normal development of meiotic nuclear divisions and spore formation.

### 3.5. Persistent Checkpoint Activity in the Absence of Cdc5 N-Terminal Domain

To gain further insight into the impact of the Cdc5 N-terminal domain in meiotic recombination checkpoint function, we also examined meiotic progression in *zip1Δ, zip1Δ cdc5-ΔN70*, and *zip1Δ cdc5-db7A* mutants by DAPI staining of nuclei. In addition, checkpoint function was monitored by following the levels of the Ndt80 transcription factor, whose production is inhibited when the checkpoint is active [46]. As expected, the *zip1Δ* single mutant exhibited a marked meiotic arrest, although a fraction of the cells resumed meiotic nuclear divisions at late time points (Figure 4A); accordingly, induction of the Ndt80/Cdc5 module was also significantly delayed compared to the wild type (Figure 3A and Figure 4B). The *zip1Δ cdc5-ΔN70* double mutant displayed an even more pronounced meiotic delay than that of *zip1Δ* (Figure 4A) that was accompanied by reduced production of Ndt80, despite the presence of higher levels of Cdc5-ΔN70 (Figure 4B). These observations are suggestive of more persistent *zip1Δ*-induced checkpoint activity when the N-terminal domain of Cdc5 is deleted. On the contrary, in *zip1Δ cdc5-db7A*, the kinetics of meiotic progression and Cdc5/Ndt80 production were comparable to those of the *zip1Δ* single mutant (Figure 4A and 4B), further corroborating that the role of the N-terminal region of Cdc5 in meiotic checkpoint control is independent of the consensus motifs for APC/C-dependent regulation.

### 3.6. The N-Terminal Domain of Cdc5 Is Required for Efficient Checkpoint Downregulation

Previous work has demonstrated that, when meiotic DSBs are repaired, Cdc5 is necessary and sufficient to silence checkpoint signaling triggering degradation of Red1, a component of the Red1-Hop1-Mek1 (RHM) complex, which is required for Mek1 activation [31]. To assess whether the more persistent checkpoint activity in the *zip1Δ cdc5-ΔN70* mutant was due to the inability of Cdc5ΔN70 to turn the checkpoint off, we took advantage of the use of a *CDC5* version controlled by the *GAL1* promoter (hereafter, *CDC5-IN*) that can be induced by the addition of estradiol in strains harboring the Gal4 transcriptional regulator fused to the estradiol receptor (Appendix A; [47]). Thus, estradiol triggers *CDC5-IN* expression independent of its normal activator, Ndt80, allowing us to explore exclusively the effect of Cdc5 or Cdc5ΔN70 on checkpoint inactivation. For this purpose, we monitored the levels of Red1 and Mek1 proteins as well as phosphorylation of Hop1-T318 and H3-T11 as readouts of Mec1 and Mek1 checkpoint kinases, respectively. We generated *zip1Δ ndt80Δ* strains containing either the wild-type *CDC5-IN* allele or the version lacking the sequence corresponding to the 1–70 N-terminal region, *cdc5-ΔN70-IN*. In this system, the checkpoint is initially activated by the lack of Zip1, but the cells remain arrested in prophase I throughout the time course, due to *NDT80* deletion, regardless of the status of checkpoint activity. This allows us to evaluate the impact of Cdc5 on the levels of checkpoint markers without interference from cell cycle progression. Twenty-four hours after meiotic induction, when *ndt80Δ* cells of the BR strain background are arrested in prophase I, estradiol was added to half of the culture, and samples were collected at successive time points for Western blot analysis. In the absence of estradiol, the checkpoint was activated in both *CDC5-IN* and *cdc5-ΔN70-IN* strains, as manifested by the accumulation of Red1 and Mek1 and high levels of Hop1-T318 and H3-T11 phosphorylation (Figure 5A, dark green and blue bars). Induction of wild-type Cdc5 resulted in the drastic disappearance of the Red1 protein and also a strong reduction in Mek1. Likewise, phosphorylation of Hop1-T318 of H3-T11 was severely decreased (Figure 5A, light green bars). In contrast, upon induction of Cdc5-ΔN70, the Red1 and Mek1 proteins displayed more stability and did not completely disappear; also, higher levels of Hop1-T318 of H3-T11 phosphorylation persisted (Figure 5A, light blue bars). The ratio of protein levels in the presence versus the absence of estradiol clearly exhibited a more rapid decrease in all checkpoint markers after induction of *CDC5-IN* compared to *cdc5-ΔN70-IN* (Figure 5B). Collectively, our results indicate that the N-terminal domain of Cdc5 is critical for dampening meiotic checkpoint signaling via downregulation of the RHM complex to facilitate the resumption of meiotic cell cycle progression. Furthermore, this novel function of the N-terminal domain of Cdc5 is independent of the canonical motifs for APC/C regulation present in this domain and does not appear to require phosphorylation of individual CDK sites.

## 4. Discussion

Multiple events in the mitotic and meiotic cell cycles are controlled by polo-like kinases. During meiosis in budding yeast, the Ndt80 transcription factor and the polo kinase Cdc5 participate in a feed-forward autoregulatory loop that triggers exit from meiotic prophase I and entry into metaphase I when DSBs are repaired and the meiotic recombination checkpoint is satisfied. A key step in this transition is the destruction of the synaptonemal complex promoted by Ndt80-dependent induction of Cdc5. Although the direct target(s) of Cdc5 in this process remains to be identified, the degradation of several SC components upon Cdc5 induction, including Red1 and Zip1, has been observed [7,8,31]. Red1 is a component of the LEs of the SC that is also crucial for checkpoint activity. In this work, we show that when the checkpoint is activated by deletion of *ZIP1*, Ndt80-independent ectopic expression of *CDC5* is sufficient to silence checkpoint signaling and we demonstrate that the N-terminal region of Cdc5 is critically required for this function.

In mitotically cycling cells, Cdc5 is degraded at the end of mitosis by APC/C-Cdh1, but deletion of the 1–70 N-terminal region protects the protein from degradation [32,33,34]. We show here that Cdc5-ΔN70 also appears to be more stable during meiosis because it can be detected throughout the whole meiotic program. In contrast, wild-type Cdc5 appears upon exit from prophase I and disappears after meiotic divisions are completed (Figure 3A). Unexpectedly, mutation of the motifs recognized by APC/C-Cdh1 has no effect on either Cdc5 levels or function, implying that additional regulatory elements must be present in the N-terminal domain of Cdc5 controlling its stability at least during the meiotic program. The relevance of the two RXXL D-boxes and the KEN motif present in Cdc5 (Figure 1A) has been also analyzed in mitotic cells using chimeric proteins in which the first 80 amino acids of Cdc5 were fused to GFP [42]. In this system, the KEN motif and the RXXL-2 D-box appeared to be important for the regulation of Cdc5 stability and nuclear localization. Thus, the control of Cdc5 levels during meiosis differs from the canonical APC/C-Cdh1-dependent regulation described in mitotic cells. Since during meiosis Cdc5 degradation is triggered by APC/C bound to the meiosis-specific regulator Ama1, instead of Cdh1 [36], we speculate that, perhaps, APC/C-Ama1 can recognize different elements of the Cdc5 N-terminal domain.

The amount of Cdc5 during the mitotic cell cycle is also controlled by CDK phosphorylation of T23 independent of APC/C. The *cdc5-T23A* mutation generates a degron that results in barely detectable levels of Cdc5 [44]. Our results indicate that this mechanism likely operates also during meiosis because a *zip1Δ* strain transformed with a high-copy plasmid expressing *cdc5-T23A* does not exhibit an increased amount of Cdc5, consistent with the fact that the Cdc5-T23A protein is highly unstable. Thus, the *zip1Δ* strain with the multicopy *cdc5-T23A* plasmid does not accumulate sufficient levels of the protein to alleviate the meiotic delay imposed by the checkpoint. Therefore, the inability of Cdc5-ΔN70 to suppress the checkpoint when overproduced cannot be attributed to the absence of this T23 CDK phosphorylation site. On the other hand, like deletion of the 1–70 terminal region, the T29A mutation, eliminating also a CDK phosphorylation site [38], appears to stabilize Cdc5 in mitotic cells [45]. However, unlike Cdc5-ΔN70, we show here that overexpression of *cdc5-T29A* in *zip1Δ* meiotic cells alleviates the nuclear divisions delay to the same extent as overexpression of wild-type *CDC5* does. This observation indicates that phosphorylation of T29 is not required for the checkpoint function of Cdc5. Moreover, although we do not detect a notable increase of Cdc5-T29A protein levels compared to wild-type Cdc5 in strains overexpressing these genes, our results are consistent with the notion that stabilization of Cdc5 per se does not cause impaired checkpoint silencing. On the other hand, phosphorylation of Cdc5 by Cdc28 at T70 in mitotic cells is important for MEN (mitotic exit network) function [43]. We observe that, like wild-type *CDC5*, overexpression of *cdc5-T70A* is capable of partially overriding the meiotic delay in *zip1Δ* cells, suggesting that this meiotic function of Cdc5 is independent of the MEN pathway. Although we have not detected a clear meiotic effect of the single mutants on CDK sites, we cannot rule out a possible effect of combined mutations.

Using strains in which we have integrated *cdc5-ΔN70* at its genomic locus, we also report here that in an otherwise unperturbed meiosis (that is, in *ZIP1* cells), the *cdc5-ΔN70* mutant exhibits a slight delay in meiotic progression, defective sporulation, and reduced viability of the meiotic products. These relatively mild phenotypes may arise from the inability of the truncated Cdc5 protein to properly perform some of the meiotic roles of the polo kinase, perhaps by impairing its interaction with a particular substrate. Alternatively, *cdc5-ΔN70* phenotypes could be also the consequence of the unscheduled regulation of Cdc5 levels throughout the meiotic and sporulation program having a deleterious effect. In line with this possibility, it has been shown that overexpression of *CDC5* from the *CUP1* promoter during meiosis leads to reduced electrophoretic mobility and premature degradation of Spo13 [48]. Spo13 is a critical regulator of centromeric cohesion and it is required for proper chromosome segregation at meiosis I [49]; in fact, *SPO13*-deficient cells undergo a single round of meiotic nuclear division and generate dyads [50]. The reduced spore viability and impaired formation of asci containing 4 spores observed in the *cdc5-ΔN70* mutant (Figure 3) may be a consequence of altered Spo13 and/or cohesin function. Alternatively, or in addition, altered Cdc5 function or levels in *cdc5-ΔN70* may also cause problems in SPB duplication [51], leading to sporulation defects because spore morphogenesis is linked to SPB dynamics and maturation [52]. Future experiments will address these possibilities. Interestingly, defective centrosome maturation during meiosis (the mammalian equivalent of yeast SPB) has been also reported in PLK1-deficient male mice [53].

Remarkably, the *cdc5-ΔN70* mutation has a more prominent impact when the meiotic recombination checkpoint is triggered (that is, in *zip1Δ* cells). The *zip1Δ cdc5-ΔN70* double mutant shows an exacerbated meiotic block, compared to *zip1Δ*, concomitant with reduced Ndt80 levels indicative of tighter or prolonged prophase I arrest. Curiously, the *zip1Δ* mutant initially arrests in prophase I, but eventually, some cells undergo meiotic divisions. In contrast, the *zip1Δ rad51Δ* double mutant displays a tight permanent block [21], suggesting that, in the *zip1Δ* single mutant, a fraction of cells repair DSBs via a Rad51-dependent intersister pathway, leading to checkpoint inactivation and resumption of cell cycle progression. We propose that the ability of Cdc5 to downregulate checkpoint activity to promote exit from prophase I is compromised in the absence of its N-terminal region, thus explaining the prolonged meiotic block in *zip1Δ cdc5-ΔN70*. To directly address whether Cdc5 N-terminal domain is required for checkpoint inactivation, we used the *CDC5-IN* estradiol-inducible system in *zip1Δ ndt80Δ* cells. Since the *NDT80* deletion prevents exit from prophase, this approach allowed us to assess the status of the *zip1Δ*-induced checkpoint without interference from indirect effects that may arise from the impact of Cdc5 levels on meiotic cell progression and/or from additional targets of the Ndt80 transcription factor also induced during prophase exit. Despite that high levels of both Cdc5 and Cdc5-ΔN70 are achieved in this inducible system, the Cdc5-ΔN70 version fails to efficiently abrogate checkpoint activity in prophase-arrested cells (Figure 5). As described, the Red1 protein rapidly disappears upon wild-type Cdc5 induction (Figure 5A, t = 27 h). Interestingly, although eventually the Mek1 protein also drastically diminishes, it still persists at 27 h, but in a largely inactive form, as revealed by the low levels of H3-T11 phosphorylation at this time point. Since Mek1 activation is achieved by consecutive *trans* and *cis* phosphorylation events that require Hop1/Red1 [16,54,55], this observation suggests that checkpoint down-regulation initially involves Mek1 dephosphorylation followed by progressive degradation of inactive Mek1. It is likely that Cdc5-induced Red1 degradation is the initial trigger for Mek1 inactivation. Red1 is subject to extensive phosphorylation at several sites, but its biological relevance is under debate [56,57]. Whether Red1 is a direct target of Cdc5 phosphorylation remains to be established. Curiously, as Red1 is degraded following Cdc5 induction, we observe a faint slow-mobility band that could be indicative of Cdc5-dependent phosphorylation (Figure 5A, red arrowhead).

Strikingly, induction of Cdc5-ΔN70 fails to efficiently silence checkpoint activity, as clearly manifested by the high levels of Hop1-T318 and H3-T11 phosphorylation that persist in the *cdc5-ΔN70-IN* mutant. Higher amounts of the Red1 and Mek1 proteins are also maintained in the absence of the N-terminal domain of Cdc5, explaining the maintenance of checkpoint activity. Curiously, Mek1 is much more stabilized than Red1 in *cdc5-ΔN70-IN* (Figure 5B). This finding suggests that relatively low amounts of Red1 are sufficient to sustain Mec1-dependent Hop1-T318 phosphorylation and the subsequent Mek1 autophosphorylation to acquire substantial levels of kinase activity to phosphorylate downstream Mek1 targets (i.e., H3-T11). This observation further supports the notion of increased protein stability for active Mek1.

The recognition of Cdc5 known substrates is usually exerted by the interaction of the polo box domain with the substrate previously primed by phosphorylation. Nevertheless, non-canonical interactions of Cdc5 with certain substrates, such as Dbf4 or Exo1, are recently emerging [58,59]. We hypothesize that, in addition to the presence of regulatory sites controlling Cdc5 stability, the N-terminal region can also contribute to facilitating the direct interaction of Cdc5 with its relevant target(s) to sustain checkpoint downregulation. In the absence of this domain, the interaction of Cdc5-ΔN70 with its substrate(s) may be less stable resulting in an impaired action despite the presence of higher amounts of the active kinase. Whether Red1 itself or other component(s) of the checkpoint signaling pathway and/or recombination machinery are the direct targets of Cdc5 in this process, will be a matter of future studies. In mice, PLK1 also drives SC disassembly promoting dissociation of SCP3, which, like yeast Red1, is a LE component [60], suggesting that the role of polo-like kinases in downregulating meiotic checkpoint activity could be also conserved. Curiously, structure analysis of both yeast Cdc5 and human PLK1 proteins using the AlphaFold Database [61] predicts well-structured kinase and polo-box domains, but it is not capable of modeling the structure of the N-terminal domain in both proteins suggesting a largely disorganized region (Appendix A). Although the amino acid sequences of the N-terminal regions of yeast Cdc5 and human PLK1 do not show homology, it is tempting to speculate that they may share a common regulatory function.

## Figures and Tables

**Figure 1 cells-10-02561-f001:**
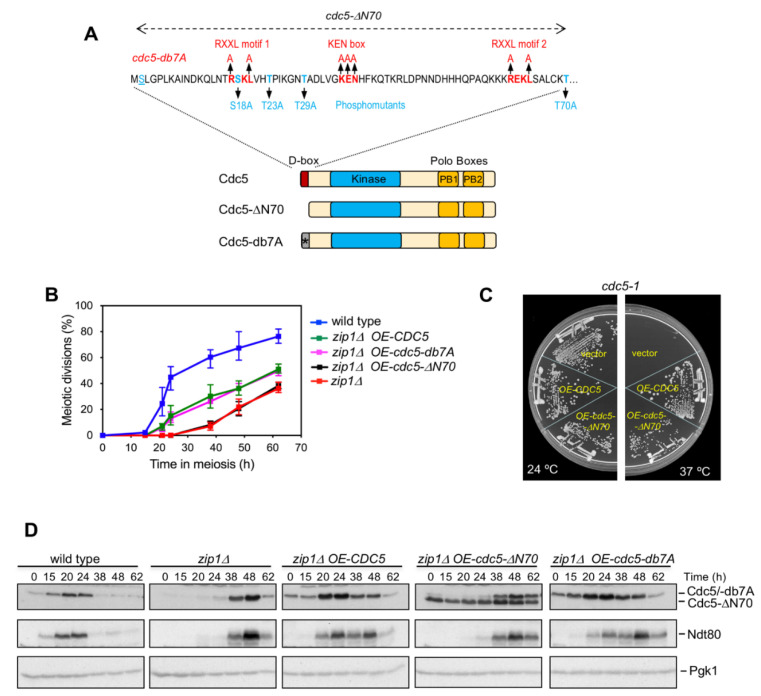
Unlike wild-type Cdc5 and Cdc5-db7A, overproduction of Cdc5-ΔN70 does not accelerate meiotic progression in *zip1Δ*. (**A**) Schematic representation of the polo-like kinase Cdc5, as well as the Cdc5-ΔN70 and Cdc5-db7A mutant versions, indicating the main functional domains: D-box, kinase, and polo-box. The sequence of the first 70 amino acids of Cdc5 is also presented. The consensus motifs RXXL and KEN for APC/C recognition are shown in red, as well as the corresponding mutations to generate *cdc5-db7A*. The reported CDK-dependent phosphorylation sites and the corresponding phosphomutants analyzed are shown in blue. (**B**) Time-course analysis of meiotic nuclear divisions; the percentage of cells containing two or more nuclei is represented. Error bars: SD; n = 6. At least 600 cells were scored for each strain at every time point. (**C**) The thermosensitive *cdc5-1* mutant was transformed with empty vector or high-copy plasmids expressing *CDC5* or *cdc5-ΔN70*, as indicated. The plate was incubated at permissive (24 °C) or restrictive (37 °C) temperature. The *cdc5-ΔN70* allele is able to complement the growth defect of *cdc5-1* at 37 °C. (**D**) Western blot analysis of Cdc5/ Cdc5-ΔN70/Cdc5-db7A and Ndt80 production throughout meiosis in the same strains analyzed in (**B**). The Cdc5 protein was detected with an anti-Cdc5 antibody. Pgk1 was used as a loading control.

**Figure 2 cells-10-02561-f002:**
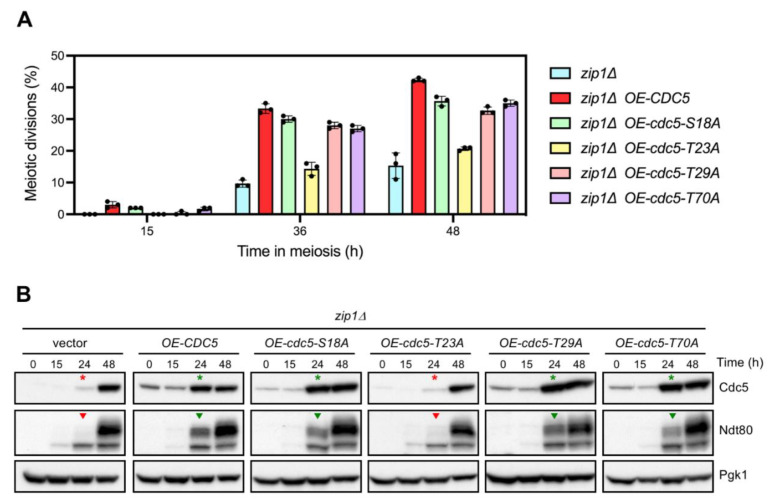
Functional analysis of identified CDK sites in the N-terminal domain of Cdc5. (**A**) Percentage of cells containing two or more nuclei at the indicated time points in meiosis. Error bars: SD; n = 3. At least 300 cells were scored for each strain at every time point. (**B**) Western blot analysis of Cdc5 and Ndt80 production throughout meiosis in the same strains analyzed in (**A**). The Cdc5 protein was detected with an anti-Cdc5 antibody. Pgk1 was used as a loading control. Asterisks and arrowheads mark Cdc5 and Ndt80 production, respectively, at the 24 h time point. See text for details.

**Figure 3 cells-10-02561-f003:**
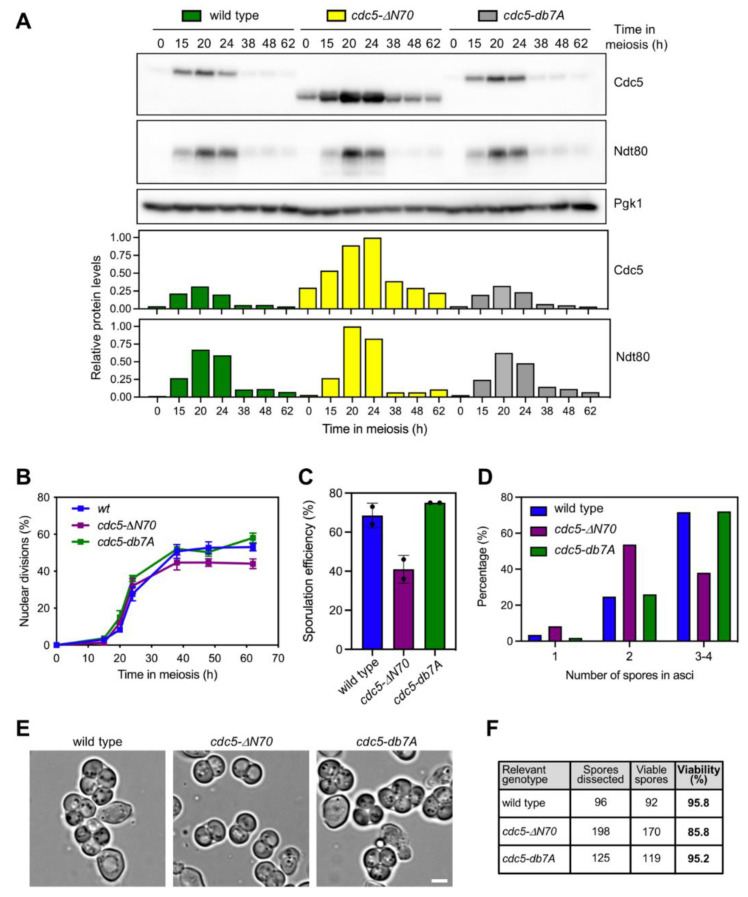
Meiotic progression, sporulation, and spore viability are affected in the *cdc5-ΔN70* mutant, but not in *cdc5-db7A.* (**A**) Western blot analysis of Cdc5 and Ndt80 production throughout meiosis. The Cdc5 protein was detected with an anti-Cdc5 antibody. Pgk1 was used as a loading control. Quantification of relative Cdc5 and Ndt80 levels is presented in the graphs. Protein levels were normalized with Pgk1 and relativized to the maximum value in the experiment set to 1. (**B**) Time course analysis of meiotic nuclear divisions; the percentage of cells containing two or more nuclei is represented. Error bars: SD; n = 3. At least 300 cells were scored for each strain at every time point. (**C**) Sporulation efficiency after 62 h in meiosis. Error bars: range; n = 2. At least 300 cells were scored for each strain. (**D**) Percentage of asci containing 1, 2, and 3–4 spores, as indicated, after 3 days on sporulation plates. (**E**) Representative bright-field microscopy images of asci from samples analyzed in (**D**). Scale bar, 2 μm. (**F**) Spore viability analyzed by tetrad/triad dissection. In the case of *cdc5-ΔN70*, only 3-spore asci were dissected.

**Figure 4 cells-10-02561-f004:**
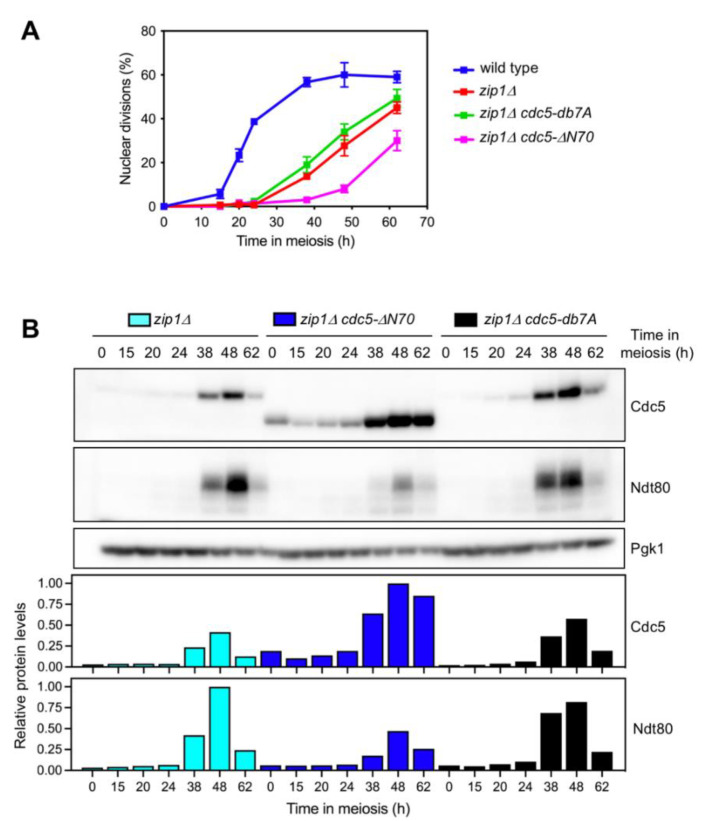
Persistent checkpoint activity in the *zip1Δ cdc5-ΔN70* mutant. (**A**) Time-course analysis of meiotic nuclear divisions; the percentage of cells containing two or more nuclei is represented. Error bars: SD; n = 3. At least 300 cells were scored for each strain at every time point. A wild-type strain was also included as a reference for the *zip1Δ*-induced delay. (**B**) Western blot analysis of Cdc5 and Ndt80 production throughout meiosis. The Cdc5 protein was detected with an anti-Cdc5 antibody. Pgk1 was used as a loading control. Quantification of relative Cdc5 and Ndt80 levels is presented in the graphs. Protein levels were normalized with Pgk1 and relativized to the maximum value in the experiment set to 1.

**Figure 5 cells-10-02561-f005:**
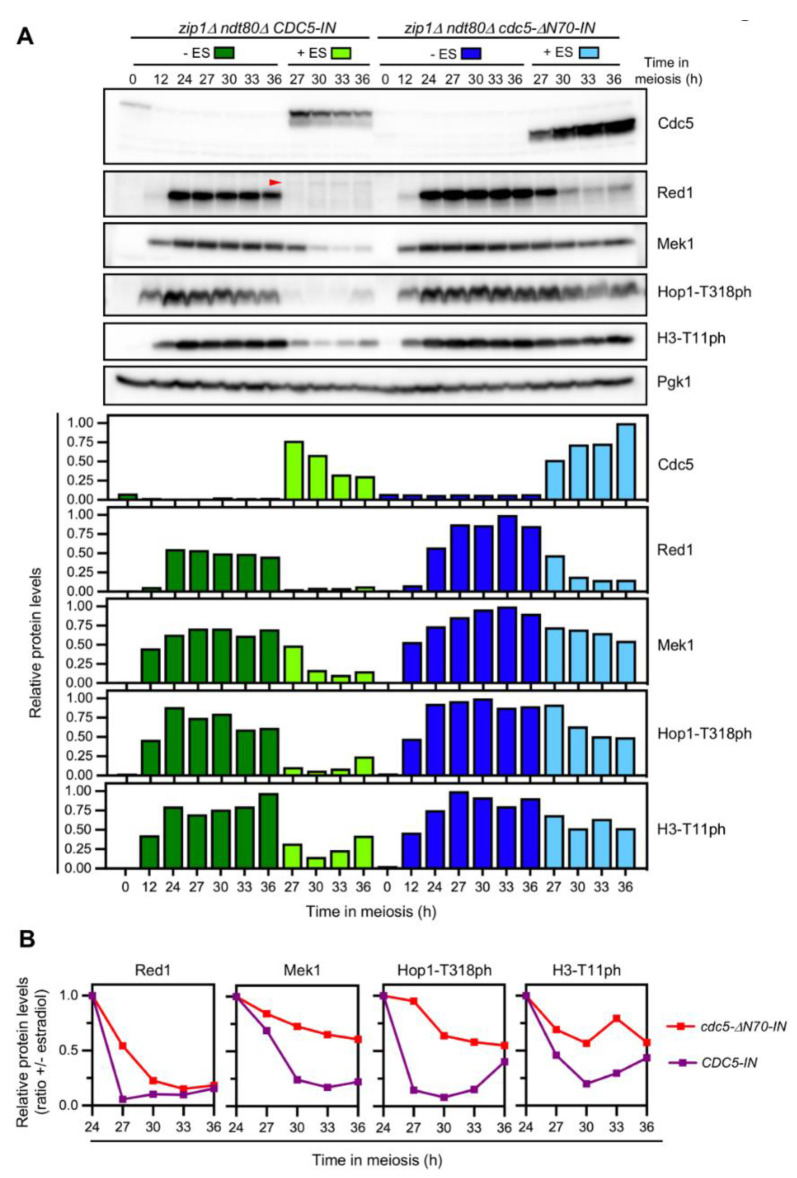
The role of Cdc5 in meiotic checkpoint termination requires its N-terminal domain. (**A**) Western blot analysis of Cdc5, Red1, and Mek1 production, as well as Hop1-T318 and H3-T11 phosphorylation (ph) as markers of checkpoint activity. Pgk1 was used as a loading control. Estradiol (ES) was added to meiotic cultures of the indicated strains 24 h after induction of meiosis. At this time, the cultures were split in two (-ES and +ES) and samples were taken at successive time points from 27 to 36 h (see Appendix A). The red arrowhead in the Red1 panel points to a putative lower mobility form of Red1 appearing upon Cdc5 (but not Cdc5-ΔN70) induction. Quantification of relative levels of the proteins and checkpoint markers analyzed is presented in the graphs. Protein levels were normalized with Pgk1 and relativized to the maximum value in the experiment set to 1. (**B**) The graphs represent the ratio of relative abundance of Red1, Mek1, Hop1-T318ph, and H3-T11ph in the presence and absence of estradiol; that is, with and without Cdc5 or Cdc5-ΔN70 induction. Note that at the latest time points (33–36 h) checkpoint activity slightly resumes in *CDC5-IN* coincident with a drop in wild-type Cdc5 levels (panel A, light green bars), likely reflecting an inherent instability of wild-type Cdc5 (but not Cdc5-ΔN70) during meiotic prophase I.

## Data Availability

All data supporting the reported results can be found within the article and the Appendix A.

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
