# Peer review of "The N-Terminal Region of the Polo Kinase Cdc5 Is Required for Downregulation of the Meiotic Recombination Checkpoint"

_cells, 2021, doi:10.3390/cells10102561_

Round 1

Reviewer 1 Report

In this article, the authors investigated the function of N-terminal domain of the polo-like kinase CdC5 in meiosis in unperturbed conditions or in conditions where the meiotic recombination checkpoint is activated. Cdc5 is well known for its many roles during meiosis, including promoting exit from the recombination checkpoint activated in mutants such as zip1.

The authors investigated the function of this region of Cdc5, because it was previously described as necessary for the degradation of Cdc5 by the APC/C complex during mitosis.

They found that the N70 domain of Cdc5 is important for the exit from, or to bypass, the recombination checkpoint, but surprisingly, the consensus sites for APC/C-mediated degradation are not important for this function.

This raises two possibilities: either it is not the destruction of Cdc5 by APC/C that is involved, or, the APC/C mediates destruction of Cdc5 through other sites than the consensus ones tested here during meiosis.

The data are interesting, and open new avenues to test the function of Cdc5 carried by this N-terminal region during meiosis, such as identifying potentially new interacting partners and substrates. The experiments are well done and the data support in general the conclusions, although there are several points that should be addressed, listed below.

Major point:

My major concern is that the paper does not demonstrate that the APC/C destruction of Cdc5 is not involved, because the point mutant of the consensus sites has not been fully characterized in mitotic cells, and even if it had, APC/C may target other sites in meiotic cells. It may be outside of the frame of this manuscript, but it would be very useful if the authors could show if the protein mutated for the RXXL KEN motifs is still degraded in mitosis, which apparently has not been properly tested in previous studies. Does this mutant, that the authors have generated for this study, have any defect in the mitotic cycle?

Likewise, it would be important to ask if APC/C still acts on the consensus sites mutant of Cdc5 during meiosis.

If these experiments are not possible, at least the authors should tone down the conclusion that the effect they see is independent of impairing Cdc5 destruction by APC/C. Line 100: could be “uncover potentially unrecognized aspects of Cdc5 function…” since we do not know from this manuscript if the function identified here is APC/C dependent or not.

Minor points:

  • Abstract line 29: “compared to the single zip1∆ mutant, the zip1∆ cdc5∆N70…”
  • Figure 1D: it would facilitate understanding to indicate, in this figure but also the other western blots of the paper, that an antibody against Cdc5 (I guess, from reading the Materials and Methods?), and not anti-HA, is used to reveal both endogenous and overexpressed Cdc5.
  • Line 254-256: for clarity, the authors should cite here, and not only in the Discussion, the fact that the T23A mutant is known (from ref 43) to have much reduced levels, likely explaining why no overexpression is obtained for this mutant.
  • Line 260 and in the Discussion: could the authors envisage that some of these three CDK sites (18, 29 and 70) are redundant? Would it be worth testing the triple mutant? This should be discussed as a possibility, or cited if this kind of mutant was already obtained to look at the mitotic functions.
  • Could the authors comment on the fact that the cells overexpressing Cdc5 from the 2 micron pRS426 already show Cdc5 signal at the 0h time point of meiosis (e.g. Fig 1D, 2B)? Was this previously observed?
  • Same question for the presence of the cdc5∆N70 behind its endogenous promoter at the 0h time-point (Fig. 3A, 4B).

Typos:

  • Line 347: “did not completely disappear”
  • Line 372: “in the mitotic and meiotic cell cycles”
  • Line 426: I guess the authors meant “when the meiotic recombination checkpoint is triggered”?

Reviewer 2 Report

The manuscript cells-1386331,"The N-terminal region of the polo kinase Cdc5 is required for downregulation of the meiotic recombination checkpoint" by González-Arranz et.al. describes experiments directed toward investigating the role of the amino-terminus of Cdc5 on regulation of meiotic checkpoint in S. cerevisiae. Key finding presented in the manuscript are:

  1. Over-expression of CDC5 allows a bypass of zip1Δ induced checkpoint delay during sporulation.
  2. Deletion of the amino-terminal 70 residues of Cdc5 yields a variant that is functional based on its ability to suppress a cdc5-1 mutant but is unable to suppress the zip1Δ-mediated checkpoint during sporulation. In contrast, point mutations that disrupt canonical D-box and ken box motifs in the amino-terminal allow bypass of the checkpoint when over expressed and behave similar to CDC5.
  3. Mutations in Cdc5 S18A, T29S, T70A all are functional when over expressed with regard to bypass of the zip1Δ checkpoint. In contrast T23A is defective in bypass, interesting T23 this matches the Cdk consensus better that the other sites.

4 The Δ70 mutant of CDC5 when integrated at endogenous locus displays a substantial increase in protein abundance and can drive an increase in Ndt80 synthesis further supporting that it has some function.

  1. The db7A mutant displays wild-type behaviour when integrated at the endogenous locus suggesting that information outside of the canonical D-box and ken motifs is influencing the protein stability and checkpoint by-pass capability.
  2. The Δ70 mutant shows delayed progression through meiosis, reduced sporulation frequency and spore viability. Interestingly the cells display a large increase in Dyad and triad asci. In contrast the db7A point mutants behave like CDC5.
  3. Expression of Δ70-Cdc5 in a zip1Δ diploid results in a more substantial checkpoint delay than expression of Cdc5 despite the increased protein abundance of the Δ70-Cdc5. The Δ70 mutant is defective in down regulation of Mek1, Red1 and phosphorylation of those proteins.

This is a clear and well written manuscript. The experiments are well controlled and data are clearly presented. The authors provide an important finding that the presence of a D-box or ken box like sequence does not necessarily mean that those elements define or even influence the stability of a protein. The results may be a meiosis-specific issue but are worth reinvestigating in cells that are mitotically proliferating.

The supplemental information is valuable and clearly presented.

Comments:

  1. Fig 2B could use a bit more explanation.

This is a western blot of extract from cells harbouring an empty vector or several versions of CDC5. Is the blot probed with anti-HA to see the tagged, over expressed protein or is it probed with anti-Cdc5 to see all the Cdc5. If only the HA tagged protein why are we seeing Cdc5 in the vector blot. If total Cdc5 then it is difficult to tell if anything is being expressed in the T23A blot. Just a clarification of what is being probed will be helpful.

  1. Similarly it would be helpful to clarify if GAL1-CDC5 and GAL1-Δ70-CDC5 are tagged. This is not included in the plasmid descriptions. Since all of the other versions of CDC5 used in the paper have an HA-tag on the N-terminal it would be helpful to know that the N-terminal tag is not contributing to the phenotypes and lack of elevated stability shown by db7A-Cdc5.

  1. Line 353 function of the N-terminal “does not appear to require phosphorylation of individual Cdk sites”. Perhaps modify this statement as Fig 2A suggests that T23, the site best matching CDK consensus is required for functional bypass of the checkpoint, at least when over expressed.
